# Just Noticeable Difference Model for Images with Color Sensitivity

**DOI:** 10.3390/s23052634

**Published:** 2023-02-27

**Authors:** Zhao Zhang, Xiwu Shang, Guoping Li, Guozhong Wang

**Affiliations:** School of Electrical Engineering, Shanghai University of Engineering Science, No. 333, Longteng Road, Songjiang District, Shanghai 201620, China

**Keywords:** just noticeable difference, perceptual redundancy, masking effect, visual saliency, color sensitivity

## Abstract

The just noticeable difference (JND) model reflects the visibility limitations of the human visual system (HVS), which plays an important role in perceptual image/video processing and is commonly applied to perceptual redundancy removal. However, existing JND models are usually constructed by treating the color components of three channels equally, and their estimation of the masking effect is inadequate. In this paper, we introduce visual saliency and color sensitivity modulation to improve the JND model. Firstly, we comprehensively combined contrast masking, pattern masking, and edge protection to estimate the masking effect. Then, the visual saliency of HVS was taken into account to adaptively modulate the masking effect. Finally, we built color sensitivity modulation according to the perceptual sensitivities of HVS, to adjust the sub-JND thresholds of Y, Cb, and Cr components. Thus, the color-sensitivity-based JND model (CSJND) was constructed. Extensive experiments and subjective tests were conducted to verify the effectiveness of the CSJND model. We found that consistency between the CSJND model and HVS was better than existing state-of-the-art JND models.

## 1. Introduction

With the development of digital media technology, the volume of image/video data is exploding. The question of how to efficiently compress image/video becomes a huge challenge. In traditional image/video compression schemes, spatial and temporal redundancies are removed according to the statistical correlation of signals, to achieve high-efficiency compression [1]. However, as the final recipient of these signals, the human visual system (HVS) has perceptual redundancies owing to its own limitations. Researchers are committed to improving processing techniques by considering the characteristics of HVS to remove as much perceptual redundancy as possible and achieve higher compression ratios without reducing perceptual quality. One popular method that measures perceptual redundancy and has been intensively researched is the just noticeable difference (JND) model.

Research has shown that the visibility of the HVS is limited for all visual signals. The human eyes can only notice changes beyond a certain threshold of visibility, and this threshold is called the just noticeable difference (JND) [2,3]. By simulating the perceptual properties of the HVS, the JND model was developed to estimate the visibility threshold of human eyes. When the change in pixel value is below this threshold, this change is not perceived by human eyes. Therefore, an appropriate JND model can accurately estimate the visibility threshold of the HVS and is frequently applied to image/video processing, such as in information concealing and watermarking [4,5,6], perceptual image/video coding [7,8], and perceptual quality assessment [9,10,11].

Pixel-domain JND models typically consider the background luminance adaption effect and masking effect [12]. The luminance adaption effect reflects the visual sensitivity of the HVS to different degrees of background luminance. On the other hand, the masking effect is a complex mechanism of perception that is the result of multiple stimuli interfering and interacting with each other. For example, image features such as texture [13], structure [14], spatio-temporal frequency [15], and pattern [16] may have an influence on the visibility threshold. It was found that the HVS is highly skilled at summarizing the regularity of images and extracting repetitive visual patterns for comprehension [17]. The complexity of visual patterns is closely related to the masking effect. Moreover, the HVS devotes more attention to edge regions [18]; the distortion around edge regions is more easily perceived than other regions. Thus, it is necessary to provide protection for edge regions to prevent obvious distortions.

Visual saliency (VS) [19] and color sensitivity (CS) are two significant perceptual characteristics of the HVS. According to the visual saliency mechanism, regions with high saliency make a greater contribution to perceptual quality. The human eyes preferentially focus on regions with high saliency, for a greater length of time. Therefore, the introduction of visual saliency is beneficial in estimating the masking effect more accurately. Moreover, the HVS has different sensitivity to different color components, and the degree of distortion that can be tolerated differs according to color. Therefore, the color sensitivities of the HVS should be accounted for in the construction of the JND model.

In this paper, we propose a color-sensitivity-based JND model (CSJND) in the pixel-domain. The performance of the CSJND model was verified through extensive experiments and subjective tests. Our main contributions are threefold:The masking effect, including contrast masking, pattern masking, and edge protection, are adaptively modulated by visual saliency, which is a more comprehensive measure of the masking effect.As the color sensitivities of the HVS to different color components differ, we applied color sensitivity modulation based on the visual sensitivity of human eyes to Y, Cb, Cr components. This is the first JND model that accounts for color sensitivity.The proposed CSJND model was utilized to guide the injection of noise into the images. Our experimental results demonstrated that the CSJND model could tolerate more noise and had better perceptual quality.

The organization of this paper is as follows: In Section 2, we provide a literature review of related work. Section 3 introduces the proposed CSJND model in detail. The experimental results and analyses are provided in Section 4. Finally, Section 5 concludes the paper and provides an outlook for future work.

## 2. Related Works

During the past two decades, plenty of JND models have been intensively researched and developed. Existing JND models can be classified into two types depending on the domain of the JND threshold that is computed. One type is the pixel-domain JND model [12,13,14], where JND thresholds are directly computed according to image pixels. The second type is the transform-domain (or subband-domain) JND model [20,21], where images are required to be converted to a transform domain. The pixel-domain JND can be considered as a compound effect of all transformed domains. In terms of practical operational efficiency, it is more convenient to calculate JND directly from pixel values without a transformation step. 

In terms of pixel-domain JND models, Chou et al. [12] constructed an early JND model using luminance adaptation (LA) and contrast masking (CM), but this model ignored the overlapping effect of these two factors. Based on this study, Yang et al. [13] designed a nonlinear additivity model for masking (NAMM) to reduce the overlapping effect between LA and CM. However, the above two JND models overestimated the JND threshold of the edge regions. Liu et al. [14] decomposed images into a structural image and a textural image, then computed the edge masking effect and texture masking effect, respectively. Chen et al. [15] developed a foveated JND (FJND) model consisting of a foveation model, spatial model, and temporal model. In [22], Wu et al. introduced the free-energy principle to determine the JND thresholds of disordered regions. Due to the distinct properties of screen content images (SCI), Wang et al. [23] developed a JND model for SCI, which could be applied to screen compression algorithms. Wu et al. [24] proposed an enhanced JND model for images with pattern complexity. Chen et al. [25] constructed an asymmetric foveated JND model using the effect of eccentricity on visual sensitivity. Wang et al. [26] proposed a superpixel-wise JND model based on region modulation. Jiang et al. [27] took a different approach and proposed a JND model based on top-down design philosophy. They utilized the KLT transform to predict a critical perceptual lossless (CPL) image from the original image, and then used the difference between the CPL and original image as the JND map. Pixel-domain JND models can be applied in image quality assessments, such as artificially added noise, distortion metrics, or used in combination with referenceless quality metrics [9].

For transform-domain JND models, Ahumada et al. [28] proposed a JND model for grayscale images by computing the spatial contrast sensitivity function (CSF). Watson [29] further considered the influence of luminance adaptation and contrast masking, and proposed the DCTune method. Zhang et al. [30] took spatial CSF, luminance adaptation, and intra- and inter-band contrast masking into account to establish the JND model. Wei et al. [21] introduced gamma correction, a block classification method, and a temporal modulation factor to improve accuracy of the JND model. Wang et al. [31] developed a Saliency-JND (S-JND) model consisting of a visual attention model and visual sensitivity model. Wan et al. [32] designed an orientation-based JND model, which determined orientation regularity according to the DCT coefficient distribution. Wang et al. [33] presented an adaptive foveated weighting JND model based on a fixation point estimation method. Transform-domain JND models are usually applied in the transform and quantization steps of video coding [21].

With the development of artificial intelligence in recent years, researchers have considered building JND models based on deep learning. Ki et al. [34] proposed two learning-based just-noticeable-quantization-distortion (JNQD) models and used them for preprocessing in video coding. Liu et al. [35] introduced a picture-wise JND prediction model based on deep learning, which could be applied to image compression. Shen et al. [36] proposed a patch-level structural visibility learning method to infer the JND profile.

However, currently existing JND models ignore the perceptual characteristic of the HVS in its varied sensitivity to different color components. For example, in the RGB color space, human eyes have different sensitivities to R, G, and B components, being the most sensitive to green, followed by red, and finally blue [37]. In the YCbCr color space, human eyes are most sensitive to the luminance component Y, followed by the chrominance components Cb and Cr [38]. Reviewing previous JND models, it is evident that they did not adequately consider the masking effect or the differences in color sensitivity to different channels. In this paper, we propose a color-sensitivity-based JND model (CSJND), which makes up for these shortcomings. To our knowledge, this is the first work that has considered color sensitivity in the establishment of the JND model.

## 3. The Proposed CSJND Model

The proposed CSJND model is composed of three main modules, which are the luminance adaptation effect (LA), visual masking effect with saliency modulation ($V\;M_{\theta }^{S}$), and color sensitivity modulation ($CS_{\theta }$). The framework of the proposed CSJND model is shown in Figure 1. Firstly, we considered the luminance adaptation effect (LA), where the visual threshold of human eyes changes according to different levels of background luminance. Then, contrast masking ($CM_{\theta }$), pattern masking ($PM_{\theta }$), and edge protection ($EP_{\theta }$) were combined to estimate the masking effect. Visual saliency was employed to adaptively modulate the masking effect, which is conducive to more accurate computations of the visual masking effect ($V\;M_{\theta }^{S}$). Next, the sub-JND thresholds $J\;N\;D_{\theta }^{S}$ of the three components Y, Cb, and Cr were calculated by the NAMM [13]. Finally, taking the sensitivities of human eyes to the Y, Cb, and Cr components into account, we added color sensitivity modulation ($CS_{\theta }$) to adjust the sub-JND thresholds of Y, Cb, and Cr components. Therefore, $C\;S\;J\;N\;D_{\theta }$ is defined as a function of $CS_{\theta }$ and ${J\;N\;D}_{\theta }^{S}$, which is expressed as
(1)$${C\;S\;J\;N\;D}_{\theta }\left(x\right)=CS_{\theta }\cdot {J\;N\;D}_{\theta }^{S}\left(x\right),$$
(2)$${J\;N\;D}_{\theta }^{S}\left(x\right)\;=\;\left\{\;L\;A\left(x\right)\;+\;V\;M_{\theta }^{S}\left(x\right)\;-\;\;\alpha \;\cdot \;\min\;\left\{\;L\;A\left(x\right)\;,\;V\;M_{\theta }^{S}\left(x\right)\;\right\}\;\right\}\;,$$
where α is the gain reduction factor due to the overlapping effect between LA and $V\;M_{\theta }^{S}$. In this paper, we set α = 0.3, which is the same as in [13]. θ represents the three color components Y, Cb, and Cr.

### 3.1. Luminance Adaptation Effect

It has been shown that HVS visibility changes according to different levels of background luminance [12]. The visibility of human eyes is limited in dark environments, but improves under better lighting conditions. As a result, the threshold of visibility differs depending on the luminance of the background. This threshold can be simulated by the luminance adaptation effect LA, which is given by
(3)$$LA\left(x\right)=\left\{\begin{array}{cc} 17\times \left(1-\sqrt{\frac{l(x)}{127}}\right)+3 & \;\mathrm{If}\;l(x)\leq 127\\ \frac{3\times (l(x)-127)}{128}+3 & \;\mathrm{if}\;l(x)>127, \end{array}\right.$$
where l(x) is the average luminance of the neighborhood where *x* is located (e.g., a 5 × 5 neighborhood).

### 3.2. Visual Masking Effect with Saliency Modulation

In this paper, contrast masking ($C\;M_{\theta }$), pattern masking ($P\;M_{\theta }$), and edge protection ($E\;P_{\theta }$) were comprehensively considered to estimate the masking effect. As there is a positive correlation between these three masking effects, we combined them by the method of multiplication, as shown in Equation (Equation 4). In addition, visual saliency was employed to adaptively modulate the masking effect to improve estimation accuracy. Thus, an improved visual masking effect with saliency modulation $V\;M_{\theta }^{S}$ was obtained, which is given by Equation (Equation 5).
(4)$$V\;M_{\theta }\left(x\right)=C\;M_{\theta }\left(x\right)\cdot P\;M_{\theta }\left(x\right)\cdot E\;P_{\theta }\left(x\right),$$
(5)$$V\;M_{\theta }^{S}\left(x\right)=V\;M_{\theta }\left(x\right)\cdot U_{S}\left(x\right),$$
where $U_{S}$ is the saliency modulation factor.

#### 3.2.1. Contrast Masking

The masking effect is weaker and the visibility threshold is lower in uniform regions with no contrast variation. On the contrary, regions with large contrast variation have stronger masking effect and higher visibility threshold. For an image *F*, the contrast variation c(x) can be calculated as the variance of pixels. The classical contrast masking function computes the visibility threshold as a linear relationship [12]. However, this method overestimates the threshold of regions with large contrast. Perceptual research has shown that the response of human eyes to light intensity change is nonlinear, and the growth rate of the visibility threshold should decrease as contrast increases [16]. Therefore, a nonlinear transducer was introduced to calculate contrast masking, which is denoted by
(6)$$CM_{\theta }\left(x\right)=0.115\times \frac{a_{1}\cdot c{\left(x\right)}^{2.4}}{c{\left(x\right)}^{2}+a_{2}^{2}},$$
where $a_{1}$ and $a_{2}$ are two control parameters. We set $a_{1}$ = 16 and $a_{2}$ = 26, which is the same as in [16]. The response map for contrast masking of the Y component is shown in Figure 2b.

#### 3.2.2. Pattern Masking

Research has shown that the complexity of a pattern is highly associated with the orientation of pixels in the pattern. Complex patterns contain more different orientations and have a stronger masking effect, whereas simple patterns have fewer orientations and their masking effect is weaker. According to [24], the orientation of pixels can be regarded as the gradient direction, which is calculated as
(7)$$\varphi \left(x\right)=\arctan\frac{g_{v}\left(x\right)}{g_{h}\left(x\right)},$$
where $g_{v}$ and $g_{h}$ represent the gradient in vertical and horizontal directions, respectively. In this paper, we use the Prewitt kernels, as shown in Figure 3.

The histogram of φ(x) is a valid representation method to describe the distribution of orientation. Therefore, the φ(x) is quantified as $\hat{\varphi }\left(x\right)$ to generate the histogram H(x), which is given by
(8)$$H_{i}\left(x\right)=\sum_{x\in R(x)}\delta \left(\hat{\varphi }\left(x\right),i\right),$$
where δ(·) is a pulse function. The histogram of a complex pattern is dense because it has many different orientations, whereas the histogram of a simple pattern is sparse. Thus, the pattern complexity $P_{C}\left(x\right)$ is computed as the sparsity of its histogram, which is given by
(9)$$P_{C}\left(x\right)=\sum_{i=1}^{N}{∥H_{i}\left(x\right)∥}_{0},$$
where ${∥\cdot ∥}_{0}$ represents the $L_{0}$ norm. The pattern masking is modeled as
(10)$$PM_{\theta }\left(x\right)=b_{1}\cdot \frac{P_{C}^{b_{2}}}{P_{C}^{2}+b_{3}^{2}},$$
where $b_{1}$ is a proportional constant, $b_{2}$ is an exponential parameter (the larger $b_{2}$ is, the faster the gain), and $b_{3}$ is a small constant. We set $b_{1}$ = 0.8, $b_{2}$ = 2.7, and $b_{3}$ = 0.1, which is the same as in [24]. The response map for pattern masking of the Y component is shown in Figure 2c.

#### 3.2.3. Edge Protection

As edge structures can attract a lot of attention, the HVS has a relatively low tolerance for the distortion near edge structures. The distortion in edge regions is more likely to be noticed by human eyes. If edge and non-edge structures are not distinguished by the algorithm, the image may appear obviously distorted. Therefore, we took edge information into account and applied protection methods for edge regions, which is calculated as
(11)$$EP_{\theta }\left(x\right)=\lambda_{\theta }\cdot G\left(x\right)\cdot W\left(x\right),$$
(12)$$G\left(x\right)=\max_{j=1,2,3,4}\left\{{\mathrm{grad}}_{j}\left(x\right)\right\},$$
where G(x) represents the maximal weighted average of gradients around the pixel at *x*. W(x) represents the edge-related weight of the pixel at *x*. $\lambda_{\theta }$ were set as 0.117, 0.65, and 0.45 for Y, Cb, and Cr components, which is the same as in [13].

#### 3.2.4. Saliency Modulation

According to the visual saliency of the HVS, regions with high saliency attract more attention from human eyes and have an important influence on the perceptual quality of an image. For regions with different visual saliencies, the masking effect is also different. Zhang et al. [39] compared several methods of saliency detection, among which the SDSP [40] had higher prediction performance and lower calculation costs. The SDSP method is expressed as
(13)$$S\left(x\right)=S_{F}\left(x\right)\cdot S_{D}\left(x\right)\cdot S_{C}\left(x\right),$$
where $S_{F}$, $S_{D}$, and $S_{C}$ denote the frequency prior, location prior, and color prior, respectively. The detailed process is described in [40].

We use Equation (Equation 14) for the normalization operation. As shown in Figure 2d, the brighter part in a saliency prediction map indicates that the value of $S^{\prime }\left(x\right)$ is closer to “1” and the saliency of this region is higher. As people pay more attention to high-saliency regions, the masking effect of high-saliency regions is weaker. In low-saliency regions, the masking effect is stronger. Visual saliency was employed to adaptively modulate the masking effect of high-saliency and low-saliency regions. The saliency modulation factor $U_{S}$ is defined as follows:(14)$$S^{\prime }\left(x\right)=\frac{S(x)-\min(x)}{\max(x)-\min(x)}.$$
(15)$$U_{S}\left(x\right)=1-S^{\prime }\left(x\right).$$

### 3.3. Color Sensitivity Modulation

Research has shown that the sensitivity of the HVS to different color components varies, and the degree of distortion that can be tolerated by human eyes to different color components also varies. Compared with the Cb and Cr components, HVS is more sensitive to distortion in the Y component. However, previous JND models have not fully accounted for this color sensitivity of the HVS. There was no difference in the processing of different color components, whether in the RGB or YCbCr space. After the sub-JND thresholds of the three components were separately calculated, they were simply combined in a ratio of $CS_{\theta }$ as 1 : 1 : 1 to establish the JND model. If color sensitivity is ignored, the JND model will not be accurate enough.

Based on the transform relationship between the YCbCr and RGB color space, Shang et al. [38] conducted extensive subjective tests of contrast sensitivity to quantify the visual sensitivity of human eyes to Y, Cb, and Cr components. This provided a theoretical basis for the development of a JND model that considers the color sensitivity of the different components. According to the test results in [38], the average ratio of the critical distance $D_{\theta }$ (θ = Y, Cb, or Cr) for Y, Cb, and Cr components that can be perceived by human eyes is 1 : 0.432 : 0.501. When the distance is fixed, the Y, Cb, and Cr components are considered to be a perceptual sub-unit of a pixel with the side length ratio of $\left(1/D_{Y}\right):\left(1/D_{Cb}\right):\left(1/D_{Cr}\right)$. Then, the percentage of each sub-unit in a unit area was computed to obtain the perceptual sensitivity parameters of Y, Cb, and Cr components, which is given by
(16)$$S_{Y}=0.695,\;S_{Cb}=0.130,\;S_{Cr}=0.175.$$

As shown in Equation (Equation 16), the perceptual sensitivity parameters of human eyes to the Y component is the largest. Evidently, the HVS is most sensitive to the Y component and distortion in the Y component is the easiest to be perceived. Therefore, the JND threshold corresponding to the Y component should be smaller. As the HVS is relatively insensitive to Cb and Cr components, a large degree of distortion in Cb and Cr components can be tolerated, so the corresponding JND thresholds can be slightly larger.

A good JND model should guide the noise to regions that are insensitive to human eyes and hide the noise as much as possible. Therefore, the color sensitivity weights in JND model should be inversely proportional to the perceptual sensitivity parameters of Y, Cb, Cr components. Taking the inverse ratio of the perceptual sensitivity parameters of Y, Cb, Cr, we obtain
(17)$$S_{Y}^{\prime }=\frac{1}{S_{Y}},\;S_{Cb}^{\prime }=\frac{1}{S_{Cb}},\;S_{Cr}^{\prime }=\frac{1}{S_{Cr}}.$$

The color sensitivity weights of three components are normalized by Equation (Equation 18). Thus, we obtain the color sensitivity weights $CS_{\theta }$, as follows
(18)$$\begin{array}{cc} \; & CS_{Y}\;=3\cdot \frac{S_{Y}^{\prime }}{S_{Y}^{\prime }+S_{Cb}^{\prime }+S_{Cr}^{\prime }},\\ \; & CS_{Cb}=3\cdot \frac{S_{Cb}^{\prime }}{S_{Y}^{\prime }+S_{Cb}^{\prime }+S_{Cr}^{\prime }},\\ \; & CS_{Cr}=3\cdot \frac{S_{Cr}^{\prime }}{S_{Y}^{\prime }+S_{Cb}^{\prime }+S_{Cr}^{\prime }}. \end{array}$$
(19)$$CS_{Y}=0.291,\;CS_{Cb}=1.554,\;CS_{Cr}=1.155.$$

The existing JND models simply combine the sub-JND thresholds of the three components at a ratio of $CS_{\theta }$ as 1 : 1 : 1 to establish the JND model, which contradicts with the perceptual properties of human eyes. In contrast to previous JND models, we assigned different weights to the three color components. The sub-JND thresholds of Y, Cb, and Cr components were adjusted according to color sensitivity. The JND map of the Y component is shown in Figure 2e. We also provided a contaminated image guided by JND noise, as shown in Figure 2f. The image (with PSNR = 27.00 dB) has almost no noticeable distortion. Through color sensitivity modulation, the threshold of the Y component was smaller, whereas the thresholds of Cb and Cr components were relatively larger. Correspondingly, the amount of noise injected into the Y, Cb, and Cr components also changed. The CSJND model guided less noise into the Y component and more noise into chrominance components Cb and Cr, which is consistent with the perceptual properties of human eyes.

## 4. Experimental Results and Analysis

In this section, we describe the extensive experiments and subjective tests conducted to evaluate the effectiveness and accuracy of the CSJND model. Firstly, we analyze the impact of the proposed factors on the JND model through comparative experiments. Then, the CSJND model is tested in comparison with other JND models to verify its performance. Finally, the correlation between the CSJND model and subjective perception is evaluated by subjective viewing tests.

In order to evaluate the performance of our model compared with different JND models, we injected random noise into each pixel of the test images, which is given by
(20)$${\hat{F}}_{\theta }\left(x\right)=F_{\theta }\left(x\right)+\beta \cdot r\left(x\right)\cdot {JND}_{\theta }\left(x\right),$$
where ${\hat{F}}_{\theta }\left(x\right)$ is the image following noise injection guided by the JND model. r(x) represents the random noise of ±1. β is a noise level controller, which makes the injected noise from different JND models have the same energy.

In the field of image quality assessment, widely used assessment metrics are the PSNR or SSIM. However, they do not provide good feedback on the subjective feelings of people, and the calculated results are sometimes inconsistent with subjective perception. VMAF (visual multi-method assessment fusion) [41,42] is an image quality assessment standard that correlates well with subjective scores. It uses a large amount of subjective data as a training set, and fuses the algorithms of different assessment dimensions by means of machine learning. In this paper, we used SSIM and VMAF to evaluate the objective quality and subjective quality of images, respectively. In addition, we also conducted subjective test experiments, and these metrics were combined to provide a more accurate assessment.

### 4.1. Analysis of the Proposed Factors

To test the influence of visual saliency and color sensitivity modulation on the performance of the JND model, experiments were conducted by the control variable method. We took the ${J\;N\;D}_{\theta }^{B}$ defined in Equation (Equation 21) as the basic model and tested the influence of the proposed factors on this basis. The JND model composed of the basic model and saliency modulation was defined as ${J\;N\;D}_{\theta }^{S}$, as shown in Equation (Equation 2). The JND model composed of the basic model and color sensitivity modulation was defined as ${J\;N\;D}_{\theta }^{C}$, as shown in Equation (Equation 22). The proposed JND model $C\;S\;J\;N\;D_{\theta }$ was composed of the basic model, saliency modulation, and color sensitivity weights modulation, as shown in Equation (Equation 1).
(21)$${J\;N\;D}_{\theta }^{B}\left(x\right)\;=\;\left\{\;L\;A\left(x\right)\;+\;V\;M_{\theta }\left(x\right)\;-\;\alpha \;\cdot \;\min\;\left\{\;L\;A\left(x\right)\;,\;V\;M_{\theta }\left(x\right)\;\right\}\;\right\}\;.$$
(22)$${J\;N\;D}_{\theta }^{C}\left(x\right)=CS_{\theta }\cdot {J\;N\;D}_{\theta }^{B}\left(x\right).$$

With the help of Equation (Equation 20), the image was contaminated under the guidance of JND models based on the different factors, including ${J\;N\;D}_{\theta }^{B}$, ${J\;N\;D}_{\theta }^{S}$, ${J\;N\;D}_{\theta }^{C}$, and $C\;S\;J\;N\;D_{\theta }$. The comparison of contaminated images from JND models based on the different proposed factors is shown in Figure 4. The contaminated images (size 768 × 512) had the same level of noise, with PSNR = 28.25 dB. However, their perceptual quality was significantly different.

As shown in Figure 4b, the basic model ${J\;N\;D}_{\theta }^{B}$ considering luminance adaptation and the masking effect showed distortion in the whole image, with a VMAF score of 80.10. As shown in Figure 4c, the model ${J\;N\;D}_{\theta }^{S}$ based on the basic model and saliency modulation was a little better. A lot of distorted areas in the wall and balcony regions were still perceived, and the VMAF score was 84.42. As shown in Figure 4d, the model ${J\;N\;D}_{\theta }^{C}$ based on the basic model and color sensitivity modulation showed significant quality improvements, with only slight distortion, and a VMAF score of 88.04. For the models based on visual saliency and color sensitivity modulation, the perceptual quality was clearly better than the basic model. In contrast, the proposed model $C\;S\;J\;N\;D_{\theta }$ had the best perceptual quality, with almost unnoticeable distortion and a VMAF score of 94.75, as shown in Figure 4e. The experimental results indicate that these two factors are beneficial in constructing a JND model that can accurately and efficiently measure perceptual redundancy.

### 4.2. Comparison of JND Models

#### 4.2.1. Performance Comparison of JND-Guided Noise Injection

For a comprehensive analysis, four state-of-the-art JND models were selected for comparison to verify the effectiveness of the CSJND model. This included Wu’s model [16] (Wu2013), Wu’s model [24] (Wu2017), Chen’s model [25] (Chen2019), and Jiang’s model [27] (Jiang2022). As the CSJND model was proposed on the basis of traditional methods, it was only compared with relevant JND models in the pixel-domain and did not involve deep learning models.

With the help of Equation (Equation 20), the image was injected with an equal level of noise under the guidance of different JND models. In the case of equal PNSR, the better the perceptual quality, the higher the agreement between the JND model and HVS. This suggests that the JND model can improve perceptual quality by guiding more noise to regions with higher visual redundancy that are insensitive to human eyes. A detailed example of the comparison is shown in Figure 5; the perceptual quality of the CSJND model was better than other JND models.

As shown in Figure 5b, the VMAF score for Wu2013 was 82.65. Wu2013 proposed that disorderly regions can hide more noise, so the model injects less noise into ordered fence regions and more noise into disordered grassland regions. However, this model overestimates the visibility threshold of disordered regions, resulting in many spots in the grassland and sky regions. As shown in Figure 5c, the VMAF score of Wu2017 was 83.41. Wu2017 introduced pattern complexity to estimate the masking effect. Although this model correctly highlights the complex grassland and sky regions, it still guides too much noise into the fence and lighthouse regions, resulting in obvious distortion. As shown in Figure 5d, the VMAF score for Chen2019 was 87.44. Chen2019 built an asymmetric foveated JND model; the fence and lighthouse regions near the center of the focus are not distorted, but the sky and grass regions away from the center are distorted. As shown in Figure 5e, the VMAF score for Jiang2022 was 90.34. Jiang2022 used the KLT transform to predict the corresponding critical perceptual lossless (CPL) image from the original image, which is different approach from the previous JND models. However, this model does not consider the visual saliency and color sensitivity of the HVS, which leads to perceptible distortions in regions such as the walls, lighthouse, and grassland. As shown in Figure 5f, our proposed CSJND model effectively improves this situation, with a VMAF score of 94.99. By introducing color sensitivity modulation to adjust the the sub-JND threshold of different components, more noise was injected into Cb and Cr components, whereas less noise was injected into the Y component. The noise is not easily perceived because human eyes are less sensitive to Cb and Cr components. Although human eyes are most sensitive to the Y component, the noise in this component was almost imperceptible due to the low level of noise.

A set of images were selected for the comparison experiment. The test images are shown in Figure 6, where I1–I6 (size 512 × 512) are frequently utilized for JND estimation [43] and I6–I12 (size 768 × 512) are frequently utilized for quality assessment [44]. The performance comparison results of noise injection are shown in Table 1. It can be seen that the CSJND model had the highest SSIM and VMAF scores under the same PSNR. The average VMAF score of the CSJND model was 9.75 higher than Wu2013, 8.14 higher than Wu2017, 5.25 higher than Chen2019, and 3.63 higher than Jiang2022. The SSIM and VMAF scores were in good agreement with the subjective perceptual quality, which demonstrates the accuracy of our data. From these experiments, we found that the CSJND model performed better in perceptual redundancy measurement than other JND models.

#### 4.2.2. Performance Comparison of Maximum Tolerable Noise Level

An effective JND model needs to hide the noise in images without reducing perceptual quality. This suggests that a JND model that can measure the perceptual redundancy more accurately can tolerate more noise. Therefore, we further tested the maximum tolerable noise level for different JND models. We used PSNR as an indicator: the lower the PNSR, the more noise the model can tolerate. The performance comparison results of maximum tolerable noise levels are shown in Table 2. It can be seen that the CSJND model had the lowest PNSR and could protect the subjective quality while hiding noise, which again demonstrates the validity of the CSJND model.

### 4.3. Comparison of Subjective Quality

Subjective viewing tests were conducted to further evaluate the performance of the CSJND model. During the experiments, we used a 24.5-inch professional-grade OLED screen monitor, SONY PVM-2541. Using Equation (Equation 20), the images were contaminated with the same level of noise under the guidance of the CSJND model and other comparison models, respectively. Then, the images contaminated by JND noise were randomly displayed side-by-side on the left or right part of the screen, in the same scene. The subjects were required to determine which image was better and how much better. If the left image was considered to be better, subjects gave a positive score; otherwise, it was given a negative score. The evaluation standard for subjective quality comparison is shown in Table 3. The subjective viewing tests were conducted in strict compliance with the ITU-R BT.500-11 standard [45].

Forty subjects with normal or corrected-to-normal vision were invited to participate in this experiment. There were 24 men and 16 women, with an average age of 24 years, and none had previous experience with image processing. The experimental results of the subjective quality comparison are presented in Table 4, where the “mean” represents the average of subjective quality scores given by subjects and the “standard deviation” denotes their standard deviation. The positive (or negative) scores indicate whether the perceptual quality of images processed by the CSJND model was better (or worse) than that of the other JND models.

Through a comparison with four other JND models (i.e., Wu2013, Wu2017, Chen2019, and Jiang2022) in subjective viewing tests, we can see that the CSJND model outperformed the other models on all the images (positive mean scores in Table 4). In three images I3, I10, and I11, in particular, there was significant perceptual quality improvement at the same level of noise (corresponding to a larger positive mean score). In addition, for the majority of the quality scores presented in Table 4, the standard deviation values (i.e., ’std’) were small, meaning that the results of quality scores given by subjects were generally consistent and the data obtained were relatively accurate. Overall, the CSJND model showed better consistency with the HVS than Wu2013, Wu2017, Chen2019, and Jiang2022 models (mean scores were 1.016, 0.952, 0.893, and 0.844 higher, respectively). Some images of the subjective viewing tests are provided in Appendix A. The CSJND model correlated well with the HVS because the color sensitivity of human eyes to different components were taken into account, leading to better performance in perceptual redundancy measurement.

## 5. Conclusions

In this paper, we propose a color-sensitivity-based JND model to measure perceptual redundancy. In contrast to previous JND models, we introduce visual saliency modulation to improve the estimation of masking effect. In addition, color sensitivity modulation based on the perceptual sensitivity of human eyes to the Y, Cb, and Cr components is performed to adjust the sub-JND thresholds of the three components. The experimental results demonstrate that the CSJND model can conceal more noise and maintain better perceptual quality with the same PSNR. In the future, we will focus on applying the CSJND model to image quality assessment and video coding.

## Figures and Tables

**Figure 1 sensors-23-02634-f001:**
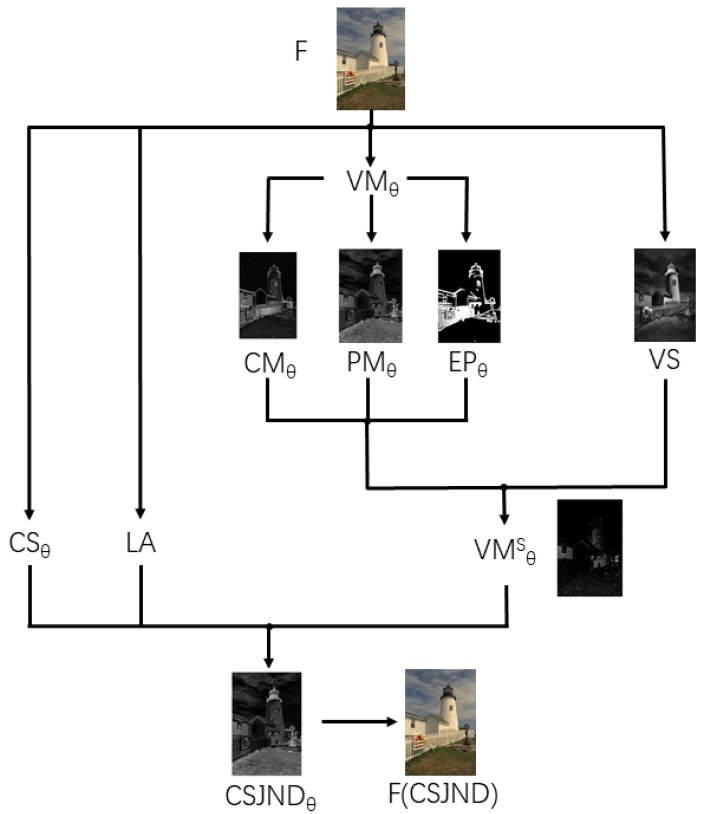
The framework of the proposed CSJND model.

**Figure 2 sensors-23-02634-f002:**
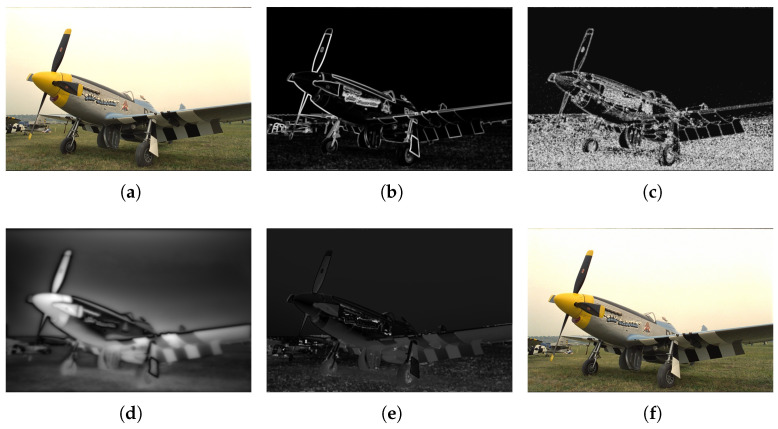
An example of JND generation and a contaminated image guided by JND noise: (**a**) the original image; (**b**) response map for contrast masking of Y component; (**c**) response map for pattern masking of Y component; (**d**) saliency prediction map; (**e**) JND map of Y component; and (**f**) JND-contaminated image, with PSNR = 27.00 dB.

**Figure 3 sensors-23-02634-f003:**
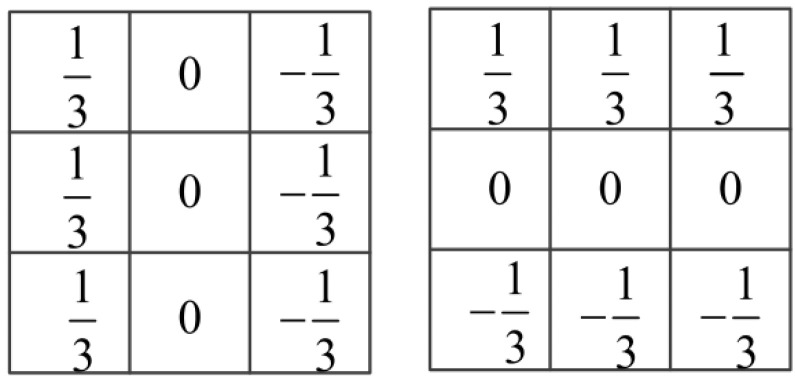
The Prewitt kernels in vertical and horizontal directions.

**Figure 4 sensors-23-02634-f004:**
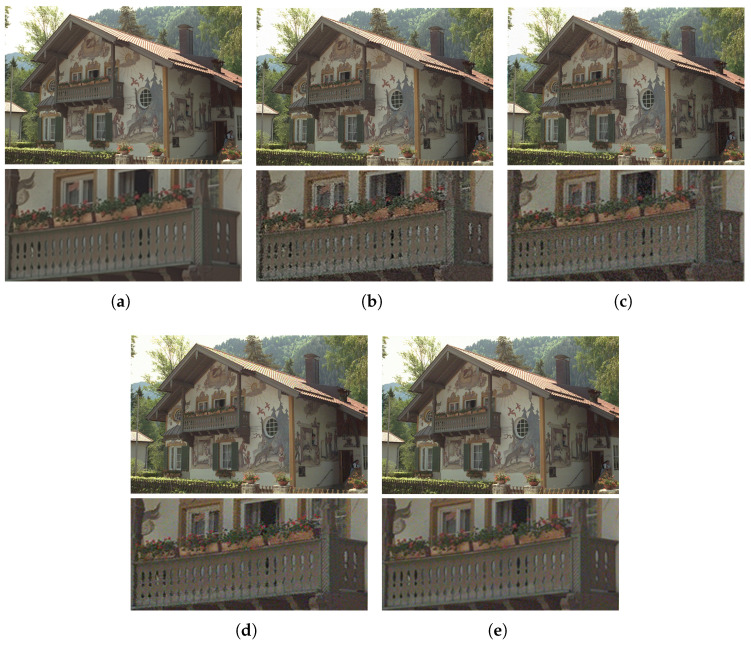
The comparison of contaminated images from JND models based on different proposed factors. The contaminated images have the same level of noise, with PSNR = 28.25 dB. (**a**) The original image. (**b**) The basic model ${J\;N\;D}_{\theta }^{B}$, VMAF = 80.10. (**c**) The model ${J\;N\;D}_{\theta }^{S}$ based on the basic model and saliency modulation, with VMAF = 84.42. (**d**) The model ${J\;N\;D}_{\theta }^{C}$ based on the basic model and color sensitivity modulation, with VMAF = 88.04. (**e**) The proposed model $C\;S\;J\;N\;D_{\theta }$, with VMAF = 94.75.

**Figure 5 sensors-23-02634-f005:**
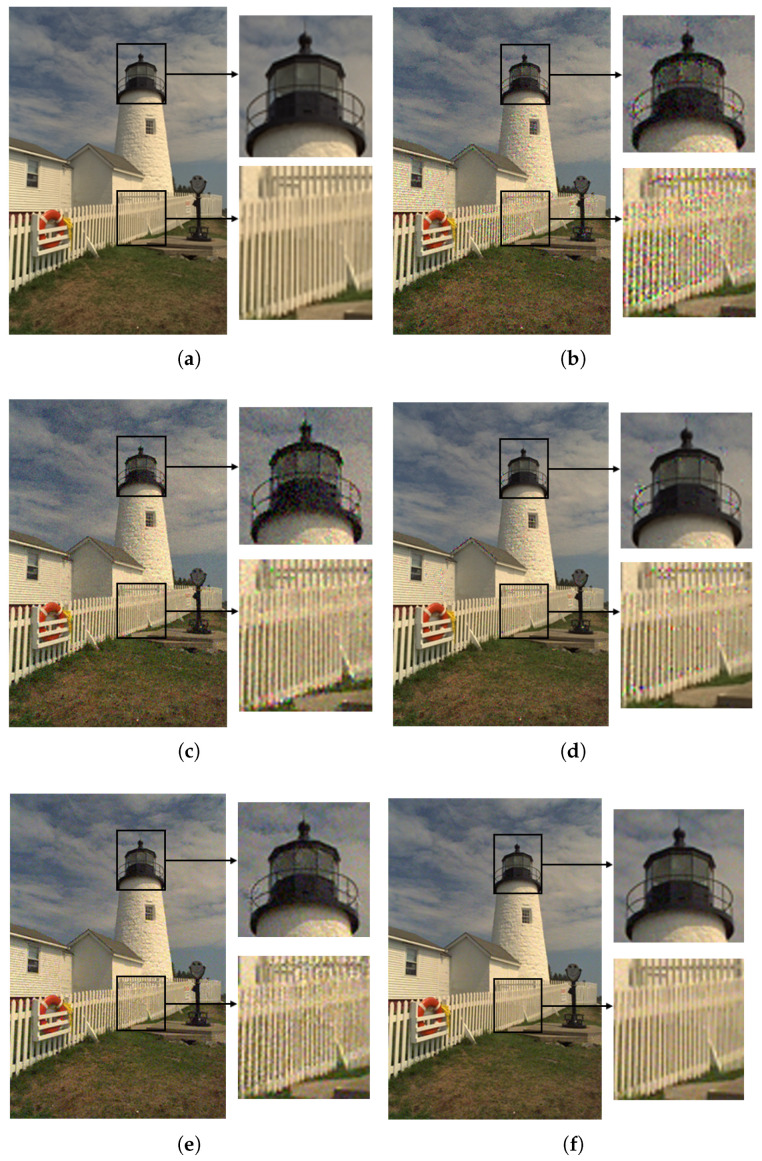
An example of the comparison of contaminated images from different JND models. The contaminated images have the same level of noise, with PSNR = 28.91 dB. (**a**) The original image; (**b**) Wu2013, VMAF = 82.65; (**c**) Wu2017, VMAF = 83.41; (**d**) Chen2019, VMAF = 87.44; (**e**) Jiang2022, VMAF = 90.34; and (**f**) The proposed CSJND model, VMAF = 94.99.

**Figure 6 sensors-23-02634-f006:**
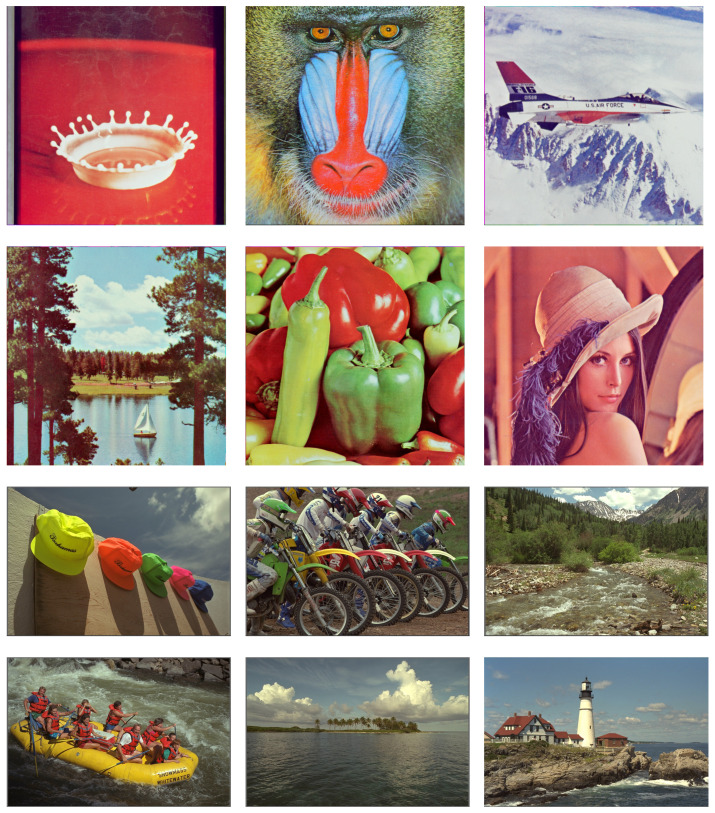
The set of test images, in order from I1–I12.

**Table 1 sensors-23-02634-t001:** Performance comparison results of noise injection based on different JND models.

Image	PSNR	Wu2013		Wu2017		Chen2019		Jiang2022		Proposed	
		SSIM	VMAF	SSIM	VMAF	SSIM	VMAF	SSIM	VMAF	SSIM	VMAF
I1	28.23	0.80	87.05	0.81	89.31	0.84	92.37	0.85	94.17	0.88	96.01
I2	24.52	0.71	78.16	0.72	80.76	0.75	83.48	0.77	85.56	0.82	90.98
I3	26.47	0.80	88.24	0.80	88.69	0.82	90.71	0.83	92.48	0.87	95.85
I4	25.95	0.74	83.33	0.75	85.04	0.78	87.34	0.82	90.02	0.85	94.50
I5	26.18	0.78	84.12	0.79	85.31	0.80	86.75	0.81	89.91	0.85	94.08
I6	26.91	0.79	86.95	0.81	87.76	0.82	88.54	0.83	91.36	0.84	95.74
I7	27.49	0.80	85.87	0.82	86.99	0.84	90.47	0.85	92.47	0.86	94.15
I8	24.91	0.73	83.42	0.79	84.64	0.81	88.40	0.82	89.40	0.84	93.03
I9	26.37	0.72	82.86	0.73	83.72	0.73	84.01	0.75	86.01	0.78	89.73
I10	24.56	0.71	80.83	0.75	85.76	0.76	88.58	0.79	90.37	0.82	94.59
I11	26.38	0.79	86.43	0.80	87.49	0.82	89.45	0.83	92.80	0.85	94.43
I12	25.16	0.73	82.82	0.75	83.88	0.77	86.79	0.79	88.94	0.84	93.91
Average	26.09	0.76	84.17	0.78	85.78	0.80	88.07	0.81	90.29	0.84	93.92

**Table 2 sensors-23-02634-t002:** Performance comparison results of maximum tolerable noise level based on different JND models.

Image	Wu2013	Wu2017	Chen2019	Jiang2022	Proposed
I1	35.72	35.28	34.45	32.56	32.25
I2	36.86	34.68	35.27	33.25	31.92
I3	35.42	36.82	35.58	34.84	32.82
I4	37.24	35.65	35.27	33.48	32.45
I5	35.54	34.78	34.82	33.65	32.28
I6	33.64	34.48	33.86	32.94	32.35
I7	36.53	35.85	33.46	34.52	32.93
I8	35.58	34.69	35.87	35.72	33.68
I9	33.64	33.93	33.41	33.85	31.38
I10	35.92	34.34	34.48	33.57	32.01
I11	34.82	35.14	34.78	34.42	32.57
I12	36.62	35.48	34.56	34.25	31.94
Average	35.63	35.09	34.65	33.92	32.38

**Table 3 sensors-23-02634-t003:** Evaluation standard for subjective quality comparison.

Description	Same Quality	Slightly Better	Better	Much Better
Score	0	1	2	3

**Table 4 sensors-23-02634-t004:** Comparison results of subjective quality.

	Methods	Proposed vs. Wu2013		Proposed vs. Wu2017		Proposed vs. Chen2019		Proposed vs. Jiang2022	
Images		Mean	Std	Mean	Std	Mean	Std	Mean	Std
I1		0.667	0.577	0.934	0.455	0.778	0.574	0.834	0.574
I2		0.762	0.539	0.952	0.540	0.836	0.650	0.752	0.458
I3		1.619	0.669	1.532	0.565	1.389	0.600	1.235	0.745
I4		0.905	0.700	0.946	0.432	0.862	0.512	0.746	0.375
I5		1.143	1.062	0.864	0.742	1.124	0.648	0.962	0.784
I6		1.190	0.750	1.183	0.790	0.854	0.820	0.943	0.824
I7		0.714	0.463	0.644	0.452	0.684	0.620	0.648	0.848
I8		0.857	1.153	0.843	0.604	0.793	0.704	0.745	0.762
I9		0.381	0.590	0.416	0.580	0.177	0.850	0.211	0.650
I10		1.333	0.658	1.348	0.694	1.368	0.569	1.248	0.480
I11		1.429	0.676	0.968	0.480	0.983	0.834	0.954	0.568
I12		1.190	0.602	0.793	0.675	0.867	0.704	0.853	0.565
Average		1.016	0.703	0.952	0.584	0.893	0.674	0.844	0.636

## Data Availability

Not applicable.

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
