# Peer review of "Just Noticeable Difference Model for Images with Color Sensitivity"

_sensors, 2023, doi:10.3390/s23052634_

Round 1

Reviewer 1 Report

The author's proposal of the JND model is very clear and with a high theoretical level of knowledge. I have no other suggestions for improvement.

Author Response

Dear reviewer,

Thank you very much for your kindly comments on our manuscript. 

Reviewer 2 Report

Authors investigate a very relevant research topic, based on the characteristics of the human eyes

can only notice changes beyond a certain threshold of visibility, and this threshold is called the just noticeable difference (JND). This topic is very useful considering different areas such as image perceptual quality assessment and image compression.

Authors stated that there are two main problems in the current JND models, which are the treating of the color components using the three channels equally, and the estimation of the masking effect is also inadequate. These problems do not permit a good correlation with the HVS, consequently, the methods fail in this regard.

Based on these deficiencies of the current JND models, authors proposed comprehensively combining the contrast masking, the pattern masking, and the edge protection to estimate the masking effect; the visual saliency of HVS is taking into account to adaptively modulate the masking effect. Authors build the color sensitivity modulation to adjust the sub-JND thresholds of Y, Cb, and Cr components. Thus, the color-sensitivity-based JND model (CSJND) is constructed.

In general, the paper is well organized and it had been well written. The methodology and results need some improvements.

The text that explain the figure 1 (proposed solution) need some improvements, for example, maybe the abbreviation need to include in the text (Luminnace adaptation → LA).

Thge masking block effect was not described in that section, at least not in a separate subsection, and it is important in the proposed solution, how the contrast, pattern, edge protection blocks are “combined” ?> Most of the block are explained but their interactions need more explanations.

In table 1 are used two image quality assessment metrics. Why were these metrics used ?

Figure 5 shows the output image of different methods, and I particularly, I did not see any difference (perceptual evaluation). Thus, a low difference between the scores obtained by SSIM and VMF (for example, I11, 0.83/0.92 and 0.85/0.94 are not noticeable by evaluators. Of course, objective metrics are necessary for these situations. In this regard, Were performed any test about the processing or memory consumption of the computers used in the tests?

In the tests, how were established the level of noise (scored by PSNR) and also the type ?

Because the ITU-R BT.500-11 standard was used. The main points of this recommendation need to be presented. Results presented in Table 4 are very positive with considerable differences of scores. I was surprised because I expected that the results of the methods, in subjective tests, would be similar. Because, in the results using objective metrics, the compared methods obtained more similar scores.

There are many relatively old references and few new/recent references, maybe authors need to improve them, including more references published in the last 5 years.

Reviewer 4 Report

This study proposes the color-sensitivity-based JND model. The idea is simple but can be sound. Overall, the manuscript is well structured. However, the paper can be improved as follows: 

- The main point of the proposed method is "color" JND that considers three color components. So, ablation study should be provided how the Color Sensitivity Modulation" contributes to the performance. In your CSJND model, compare between your models with "color sensitivity modulation" without it. 

- Figures 4 and 5 should be improved. It is hard to visually compare the images. You may magnify (zoom in) some regions to highlight the differences. 

- More visual results should be provided in a supplementary file. 

- Many typos and English writhing should be improved. 

Round 2

Reviewer 2 Report

Authors improve the manuscript, and it an be accepted in its current version.

Author Response

Thank you very much for your kind comments, your suggestions are highly beneficial to the improvement of our manuscript.

Reviewer 3 Report

Most of my concerns have been solved. It should be better to add some discussions and relations between the NR-IQA [1] and the limitations of the proposed work.

[1] Referenceless quality metric of multiply-distorted images based on structural degradation.

Author Response

We have cited reference [1] and added a discussion between NR-IQA [1] and the proposed work in Section 2.

[1] Referenceless quality metric of multiply-distorted images based on structural degradation.